# Study of Coagulation Disorders and the Prevalence of Their Related Symptoms among COVID-19 Patients in Al-Jouf Region, Saudi Arabia during the COVID-19 Pandemic

**DOI:** 10.3390/diagnostics13061085

**Published:** 2023-03-13

**Authors:** Heba Bassiony Ghanem, Abozer Y. Elderdery, Hana Nassar Alnassar, Hadeel Ali Aldandan, Wajd Hamed Alkhaldi, Kholod Saad Alfuhygy, Mjd Muharib Alruwyli, Razan Ayed Alayyaf, Shoug Khaled Alkhalef, Saud Nahar L. Alruwaili, Jeremy Mills

**Affiliations:** 1Department of Clinical Laboratory Sciences, College of Applied Medical Sciences, Jouf University, Sakaka 72388, Saudi Arabia; 2Regional Laboratory, Ministry of Health, Al-Jouf, Sakaka 72345, Saudi Arabia; 3School of Pharmacy and Biomedical Sciences, Portsmouth PO1 2DT, UK

**Keywords:** COVID-19, coagulation disorders, D-dimer, SARS-CoV-2, symptoms

## Abstract

Introduction: The coronavirus (COVID-19) has affected millions of people around the world. COVID-19 patients, particularly those with the critical illness, have coagulation abnormalities, thrombocytopenia, and a high prevalence of intravascular thrombosis. Objectives: This work aims to assess the prevalence of coagulation disorders and their related symptoms among COVID-19 patients in the Al-Jouf region of Saudi Arabia. Subjects and methods: We conducted a retrospective study on 160 COVID-19 patients. Data were collected from the medical records department of King Abdulaziz Specialist Hospital, Sakaka, Al-Jouf, Saudi Arabia. The socio-demographic data, risk factors, coagulation profile investigation results, symptom and sign data related to coagulation disorders, and disease morbidity and mortality for COVID-19 patients were extracted from medical records, and the data were stored confidentially. Results: Males represented the highest prevalence of COVID-19 infection at 65%; 29% were aged 60 or over; 28% were smokers; and 36% were suffering from chronic diseases, with diabetes mellitus representing the highest prevalence. Positive D-dimer results occurred in 29% of cases, with abnormal platelet counts in 26%. Conclusion: Our findings confirm that the dysregulation of the coagulation cascade and the subsequent occurrence of coagulation disorders are common in coronavirus infections. The results show absolute values, not increases over normal values; thus, it is hard to justify increased risk and presence based on the presented data.

## 1. Introduction

In December 2019, Chinese authorities in Wuhan identified atypical pneumonia clusters of unclear cause. On January 7, 2020, they identified a new Coronaviridae virus as the cause of the respiratory infection outbreak, naming it SARS-CoV-2 (severe acute respiratory syndrome coronavirus 2) [1]. COVID-19 has been a global health disaster, with almost 497,960,492 cases of COVID-19 and 6,181,850 deaths associated with COVID-19 documented globally [2].

Due to the abundance of viruses that bind to the angiotensin-converting enzyme 2 (ACE2) receptor in the human body, COVID-19 has the ability to impact several organs and systems in the body, including the pulmonary, central nervous, cardiovascular, hematological, urogenital, and gastrointestinal systems. In addition, lung fibrosis may indirectly injure other organs by impairing oxygen delivery and triggering a cytokine storm, which leads to the malfunctioning of immune responses, impaired coagulation, and inflammatory cell infiltration [3]. The severity of COVID-19-induced pneumonia was attributed to a number of factors, such as age, chronic obstructive pulmonary disease (COPD), smoking history, and respiratory failure [4]. COVID-19 patients who experience severe symptoms, especially those who have coexisting conditions such as chronic diseases, may quickly develop acute respiratory distress syndrome (ARDS) as well as pneumonia, with a higher mortality rate days after the disease first appears, suggesting that it may lead to a multisystem condition [1]. Fever, cough, dyspnea, wheezing, and excessive mucus production were the most prevalent symptoms of COVID-19 disease [5]. Death in severe COVID-19 infection is generally caused by the development of ARDS, sepsis, and multiple organ failure, all of which are a result of dysfunctional immunological, endothelial, and coagulation responses characterized by thrombocytopenia, leukopenia, hypercoagulation, and elevated D-dimer levels [6].

Neurological clinical manifestations of COVID-19 are generally non-specific for the SARS-CoV-2 virus. However, SARS-CoV-2 has the capability to gain direct access to the nervous system, increasing the risk for even more serious neurological complications, including ischemic stroke [7]. The most common neurologic manifestations of COVID-19 are alterations in taste and smell, headache, disorders of consciousness/cognition, and neuropsychiatric manifestations; rarer manifestations include seizures, transverse myelitis, Guillain-Barre syndrome, rhabdomyolysis, and cranial nerve palsy [8]. In individuals with pre-existing neurological disorders like Parkinson’s disease or dementia, COVID-19 may raise the likelihood of exacerbating neurological complications and vice versa [9].

Severe COVID-19 seems to have a strong link with coagulopathy and platelet count abnormalities [10]. Many individuals have coagulation abnormalities that match those of other systemic coagulopathies associated with severe infections, such as disseminated intravascular coagulation (DIC), venous thromboembolism (VTE), and/or thrombotic microangiopathy [11]. Alterations caused by COVID-19 coagulopathy, such as increased blood coagulation, result in a reduction of platelets, which leads to DIC that rarely occurs [12]. The disruption in the synthesis of coagulation proteins may result in the formation of exhaustion factors, which can lead to bleeding. Even if DIC in certain individuals is uncommon, assessing sepsis-induced coagulopathy is extremely useful in predicting the severe consequences of COVID-19 [12].

The mechanisms that trigger coagulation and exacerbate coagulative disorders in association with COVID-19 infection have been connected to immune system responses, specifically the pro-inflammatory mediator release that interacts with platelets, activating tissue factor expression, and triggering stimulation of plasminogen activator inhibitor-1, which leads to the inhibition of the fibrinolytic system and leads to endothelial dysfunction, causing thrombogenesis. Elevated D-dimer levels have been identified as a reliable biomarker for poor prognosis [13].

The aim of this study is to assess the prevalence of coagulation disorders and their related symptoms among COVID-19 patients in the Al-Jouf region of Saudi Arabia.

## 2. Materials and Methods

The study design in this research was a retrospective study on COVID-19 patients, conducted in the Al-Jouf region of Saudi Arabia. Study duration and sample size were conducted (12/2021–02/2022), with samples comprised of the medical records of 160 COVID-19 patients previously admitted (01/2020–10/2021) to isolation and intensive care units in King Abdulaziz Specialist Hospital, Sakaka, Al-Jouf, Saudi Arabia. Exclusion criteria were pregnancy, recurrent miscarriage, history of stroke or deep venous thrombosis (DVT), long history of birth control pill use and/or hormone replacement therapy, family history of coagulation disorders, and recent surgery with prolonged bed rest. For data collection and ethical approval, data were collected from the medical records department in King Abdulaziz Specialist Hospital, Sakaka, Al-Jouf, Saudi Arabia, with confidentiality, and stored securely in accordance with the study protocol approved by the Research Ethics Committee, Qurayyat Health Affairs, Reg. No.: H-13-S-071.

Sociodemographic data and risk factor information, such as smoking and history of chronic diseases, were collected for COVID-19 patients, along with results for thrombosis risk, including platelet count, PT (prothrombin time), aPTT (activated partial thromboplastin time), INR (international normalized ratio), and D-dimer level. A control group was added to the analysis for coagulation values, platelet counts, and D-dimer using data gathered from 160 normal cases without COVID-19 infection.

Symptoms and signs for the 160 COVID-19 patients, related to coagulation disorders, disease morbidity, and mortality data, were also stored. The relative risk and odds ratio at 95% CI (confidence interval) for the thrombosis risk factor in COVID-19 patients were estimated.

Data analyses were carried out using SPSS 23 statistical software, and results were analyzed using the Chi-square test. Relative risk, odds ratio, and Fisher’s exact test were used, and any differences were considered statistically significant at *p* < 0.05.

## 3. Results

A total of 160 COVID-19 patients were involved in this study. Table 1 displays sociodemographic data features in COVID-19 patients analyzed by using the Chi-Square test. A total of 104 (65%) of cases were males infected with COVID-19, while 56 (35%) were females (*p*-value < 0.001 **). Patients ranging from 19–59 years old were 113 (71%), while patients 60 years old or more were 47 (29%) (*p*-value < 0.001 **). A fraction (79%) of patients were hospitalized in isolation units and 21% in ICUs (*p*-value < 0.001 **). Table 1 also illustrates risk factors in COVID-19 patients’ prevalence. A total of 42 (28%) cases were smokers, and 118 (72%) were non-smokers (*p*-value < 0.001 **). A total of 57 (36%) cases were suffering from chronic diseases and 103 (64%) were not (*p*-value < 0.001 **).

Figure 1 shows the prevalence of chronic diseases in COVID-19 patients (36% of the total number of COVID-19 patients), with the highest percentage being that of diabetes mellitus (39%), followed by hypertension (30%), COPD (14%), chronic liver diseases (10%), and chronic renal diseases (7%) (*p*-value <0.001 **).

Table 2 displays the results of investigations into thrombotic risk and coagulation disorders. Platelet count results were normal in 74% of cases, low in 18%, and high in 8% (*p*-value <0.001 **). For coagulation profile results, PT and INR results were normal in 55% of cases, high in 34%, and low in 11% (*p*-value <0.001 **). Moreover, aPTT results were normal in 64% of cases, high in 28%, and low in 8% (*p*-value <0.001 **). D-dimer results were negative in 71% of cases and positive in 29% of cases (*p*-value <0.001 **). Table 3 demonstrates the prevalence of symptoms and signs in COVID-19 patients’ infections with cough presented the highest prevalence (64%), followed by fever (56%), muscle ache (41%), chest pain (30%), dyspnea (29%), sore throat (19%), epigastric pain (7%), and diarrhea (5%).

Table 3 displays the comparison of platelet count, coagulation profile, and D-dimer results in the control group and COVID-19 patients using an independent T test. Platelet count results were lower in COVID-19 patients than in the control group (*p*-value 0.001 *). For coagulation profile results, PT, aPTT, and INR results were higher in COVID-19 patients than in the control group (*p*-values <0.001 **, <0.001 **, and 0.006, respectively). D-dimer results were also higher in COVID-19 patients than in the control group (*p*-value 0.005 *).

Table 4 demonstrates the prevalence of symptoms and signs in COVID-19 patients’ infection, with cough presenting the highest prevalence (64%), followed by fever (56%), muscle ache (41%), chest pain (30%), dyspnea (29%), sore throat (19%), epigastric pain (7%), and diarrhea (5%).

Table 5 displays the prevalence of symptoms and signs related to precipitating thrombotic risk in COVID-19 patients, with pulmonary thrombosis (29%) having the highest percentage of symptoms and signs, followed by DVT (22%).

Figure 2 shows the prevalence of disease morbidity in COVID-19 patients and the different complications that occurred in the COVID-19 patients in the studied group. Pneumonia affected 30% of all cases, 12% had thrombosis, 5% were unconscious, 4% developed bleeding disorders, 4% had septic shock, 3% had electrolyte disturbances, and 2.5% had renal failure. COVID-19 patients who needed O_2_ therapy were 76%, and 7% needed mechanical ventilators, and the mortality rate was 2% (*p*-value <0.001 *).

Figure 3 shows types of thrombosis in COVID-19 patients with a prevalence complicated by thrombosis, with the highest percentage for DVT (53%), followed by pulmonary (26%), cerebral (16%), and cardiac thrombosis (5%), with a significant *p*-value of 0.03 *.

Table 6 shows the relative risk (RR) and odds ratio (OR) for thrombosis risk factors. Patients with chronic diseases were at the greatest risk, with an OR of 18.04 (3.65–82.4) and a RR of 13.55 (3.2–57.17). Thrombosis was associated with a high death rate, with an OR of 15.4 (1.3–179) and a RR of 5.8 (2.3–14.4). Other significant risk factors are age 60 years or more, with an OR of 12.77 (3.96–41.2) and a RR of 9.02 (3.2–25.7). Positive D-dimer was an important risk factor that had an OR of 9.16 (3.07–27.3) and a RR of 6.7 (2.57–17.6), followed by unconsciousness with an OR of 8.5 (1.9–37.3) and a RR of 4.75 (2.06–10.9), high PT and high INR with an OR of 6.8 (2.3–20.2) and a RR of 5.3 (2.03–14.07), low platelet count with an OR of 5.78 (2.08–16.06) and a RR of 4.2 (1.9–9.5), and high aPTT with an OR of 5.6 (2.04–15.4) and a RR of 4.38 (1.8–10.4). Smoking is also a risk factor with an OR of 4.88 (1.8–13.18) and RR of 3.86 (1.67–8.9), in addition to respiratory failure and mechanical ventilator with an OR of 4.75 (1.25–18.02) and a RR of 3.39 (1.37–8.4).

## 4. Discussion

COVID-19 surfaced in Wuhan, China, and rapidly spread across the globe, causing a pandemic that crippled life as we know it and put tremendous pressure on governments and individuals alike. The SARS-CoV-2 strain is incredibly contagious, and the hospitalization rate increases significantly in patients aged ≥50 and in those with underlying conditions such as hypertension or obesity [14]. Furthermore, hospitalized patients suffering from severe respiratory or systemic ailments are more susceptible to developing venous thromboembolism and thrombotic complications, which can be life-threatening [15]. Our study focused on the prevalence of coagulation abnormalities and symptoms associated with COVID-19 in patients in the Al-Jouf region.

The first COVID-19 case in Saudi Arabia was reported on March 2, 2020 [16]. There were 507,000 confirmed COVID-19 cases and 8048 fatalities as of July 19, 2021 [17]. In May 2021, Saudi Arabia documented the first case of the delta variant, and delta became the predominant variant of concern from May through June 2021. Sequenced samples revealed that delta was the most prevalent variant (at 40.9%), which, compared to the original strain, is more contagious and causes more serious symptoms [18]. Additionally, the second most abundant strains are beta (15.9%) and alpha (11.6%); this is consistent with transmission rates reported from other nations, such as the UK and France, the latter of which in June 2021 reported a rapid spread of delta [19]. Since our data were collected from January 2020 to October 2021, we speculate the aforementioned strains were circulating from May 2021 until the end of the study, in addition to the original strain that had been circulating in Saudi Arabia since the beginning of 2020.

This research found that the rate of COVID-19 infection in the Al-Jouf region was significantly higher in males (65%) than females (35%). Similarly, a study conducted in Saudi Arabia revealed that men constituted 71% of all cases [20]. On the other hand, Chinese research demonstrated that male and female patients exhibited similar susceptibility to the virus [21]. In addition, the gap in infection rates between sexes in Saudi Arabia does not comport with several studies from China, the USA, and Europe that showed a comparable number of cases in both men and women [22,23,24]. Moreover, during COVID-19 outbreaks, women in Saudi Arabia had better knowledge of infection control practices and were more compliant with the WHO prevention protocols [25]. This may explain the difference in gender infection rates in Saudi Arabia when compared with the rest of the world. Another possible explanation is that most women in Saudi Arabia tend to wear a cloth niqab, which works roughly as a mask and may prevent face touching and limit the spread of droplets.

Coughing is the most common symptom in our study, followed by fever, muscle ache, chest pain, dyspnea, sore throat, and epigastric pain. The highest prevalence of symptoms associated with (and possibly precipitating) thrombotic risk was found in pulmonary thrombosis (29%), followed by DVT, cerebral, and urinary tract thrombosis. Monitoring symptoms may assist clinicians in identifying individuals with bad prognoses, especially in high-risk patients. By promptly identifying risk factors, we can focus on high-risk groups to hopefully reduce the severity and mortality of the disease through timely diagnosis, isolation, and treatment. With this in mind, our study demonstrated that the most prevalent risk factor is diabetes mellitus (39%), followed by hypertension (30%), COPD (14%), chronic liver diseases, and chronic renal disorders. Additionally, 28% of all cases involved smokers, and 36% had chronic diseases. Other risk factors include unconsciousness, septic shock, susceptibility to bleeding, electrolyte imbalance, and renal failure. Finally, 76% of patients in this study required O2 therapy. These risk factors are associated with the progression of pneumonia in COVID-19 patients and may increase the severity of the disease [4].

We found that the most common morbidity associated with COVID-19 was pneumonia (30%), followed by thrombosis (12%). Similar findings have linked several morbidities to COVID-19, such as pneumonia with ARDS, thrombosis, renal failure, anorexia, and digestive issues [26]. Furthermore, we found that DVT was the most common type of thrombosis in COVID-19 patients, accounting for 53% of all cases, followed by pulmonary (26%), cerebral (16%), and cardiac thrombosis (5%). A study by Hanff and co-workers indicated that the most common thrombotic events in COVID-19 are pulmonary and DVT, followed by cerebral and cardiac thrombosis, which result in ischemic stroke [27]. These findings support our results that COVID-19 is highly prothrombotic. Finally, since COVID-19 infection is commonly associated with hypercoagulability, the chance of developing DVT or a potentially fatal pulmonary embolism is significantly higher as a result of small thrombus migration [28,29].

In this study, about one out of five patients (21%) was admitted to the ICU. Of these, 7% required artificial ventilation for respiratory system failure. Additional complications found in the ICU included shock and organ failure. Although ICU admissions are at 21% in Al-Jouf, the mortality rate is only 2%. This finding is in line with the global mortality rate of around 2% [30]. Numerous variables, such as age, smoking, and illness, have a significant impact on the death rate [31]. The fatality rate raises significantly in adults above the age of 50; patients above 80 years of age have a mortality rate of 14.8%; and those between ages of 70 and 79 have a mortality rate of 8%. Finally, the mortality rate at 60–69 years of age is 3.6% [32].

In our study, the risk of thrombosis substantially increased in the presence of chronic diseases, old age, positive D-dimer, thrombocytopenia, and smoking. Our coagulation panel revealed that COVID-19 patients exhibited elevated D-dimer levels in 29% of cases, elevated PT and INR in 34% of cases, and 28% had high aPTT. Both low and high platelet counts were noted. These findings are in line with the findings of other studies, which reported that coagulopathy was a common symptom of SARS-CoV-2 infection, with an increase in D-dimer, fibrinogen, PT, aPTT, and moderate thrombocytopenia [28,33]. As we have previously demonstrated, the levels of PT were significantly higher in COVID-19 patients, and this can be used as a key coagulation marker that may help us better predict the outcome of the disease, as higher PT results have been associated with an increased ICU admission rate as well as increased mortality [5]. Additionally, Zhang et al. discovered that elevated D-dimer levels above 2.0 mg/L might predict mortality with a sensitivity of 92.3% and a specificity of 83.3% [34]. Similarly, Poudel et al. found that D-dimer levels exceeding 1.5 μg/mL predict mortality in COVID-19 patients with high specificity and sensitivity [35]. Tang et al. demonstrated that elevated D-dimer, thrombocytopenia, and delayed prothrombin time have been linked to poor prognosis in patients with COVID-19 [28]. Yet, if a substantial thrombus develops in the body but is not cleared (a far more severe situation for the body), the spike in D-dimer could be moderate. In particular, even in extreme situations such as death, the rise in D-dimer is relatively moderate in suppressed-fibrinolytic-type DIC induced by sepsis [36]. Thus, to properly identify and forecast the outcome of COVID-19-associated coagulopathies, we should not depend exclusively on D-dimer as a coagulation indicator; other coagulation makers should be included.

Our analysis of COVID-19 patient data showed significantly higher (8%) and, in some cases, lower (18%) levels of platelet count. This finding may suggest that platelet abnormalities may be predictive of poor prognosis since low platelet count is associated with a fivefold increase in the risk of disease severity, which could be attributed to secondary infections [37,38]. Furthermore, only 8% of ICU patients had platelet counts of less than 100 × 10^9^/L when they were admitted, according to Wu et al. [5]. Other studies have observed the reduction and elevation of platelet count in COVID-19 patients; this phenomenon may point to an increased inflammatory state, with a low platelet count suggesting excessive platelet consumption via thrombi formation but an increase in platelet count suggesting a cytokine storm [39]. Different mechanisms can dictate viral-platelet interactions depending on the virus type; these interactions may change platelet number and function. For example, suppressing platelet production or enhancing platelet destruction, anti-viral antibodies that cross-react with platelet surface integrins, inducing systemic inflammation, and clearing of activated platelets via splenic/liver macrophages and/or phagocytosis by neutrophils may increase platelet counts in COVID-19 patients, and activated platelets may, in turn, contribute to lung injury [40].

Long-term COVID is a chronic, frequently disabling illness that affects many organ systems and is present in at least 10% of severe COVID-19 infections. It has more than 200 known manifestations. Furthermore, numerous pathogenic theories have been put forth, such as SARS-CoV-2 reservoirs remaining in tissues, immunological dysregulation, effects of SARS-CoV-2 on the microbiota (i.e., the virome, autoimmune responses, and immune system priming caused by molecular mimicry), microvascular blood clotting with endothelial dysfunction, and dysfunctional signaling in the brainstem and/or vagus nerve. Its symptoms include postural orthostatic tachycardia syndrome, dysautonomia, and myalgic encephalomyelitis/chronic fatigue syndrome [41]. The contradictory presence of both lymphopaenia and an autoimmune state in long-term COVID is puzzling; autoantibodies are directed against ACE2, ACE1, beta-adrenergic receptors, muscarinic cholinergic receptors, and a variety of self-antigens. The presence of CD4+ T-cell lymphopenia was found to be a reliable predictor of severity and hospitalization in COVID-19 individuals. This phenomenon suggests that the effects of SARS-CoV-2 may be selectively targeting CD8-expressing T lymphocytes while leaving the B lymphocytic system unaffected [42].

Coagulopathies resulting from a COVID-19 infection are possibly caused by numerous underlying mechanisms. Firstly, viral infections induce a systemic inflammatory response that alters the homeostatic balance between procoagulant and anticoagulant activities through various mechanisms, for example, endothelial dysfunction, increased von Willebrand factor, TLR (Toll-like receptor) activation, and tissue-factor pathway activation. These pathways can lead to excessive activation of the coagulation cascade, increasing D-dimer levels [43]. Additionally, during infection, levels of pro-inflammatory cytokines such as IL-1 (interleukin) and IL-6 increase as a result of the virus binding to TLR, which eventually leads to inflammation, fever, and lung fibrosis [44]. Furthermore, because of the strong expression of the virus-binding ACE2 receptor, it is widely established that the gastrointestinal system is actively involved in COVID-19 pathogenesis. The intestinal invasion of SARS-CoV-2 could disturb intestinal homeostasis and host immunological homeostasis, which were responsible for COVID-19′s adverse consequences [45,46]. The dispersion of the gut microbiota and bacteria, which is associated with socioeconomic status and could correlate with viral invasion and the severity of COVID-19 illness. During a COVID-19 infection, antibiotics are administered to treat secondary infections; this could potentially worsen the prognosis due to the decline in human microbiota that promotes the host’s immune system as a result of excessive antibiotic consumption [47]. The capacity of microbial pathogenic components to translocate via the leaky gut into the circulatory system allows inflammatory cytokines to be secreted by triggering pattern recognition receptor-like TLRs and NOD-like receptors, resulting in systemic inflammation. Moreover, the amounts of various microbial species varied in tandem with the course of COVID-19 and were linked to biomarkers of host immunology and inflammation [48]. An additional possibility for the increased inflammation in COVID-19 patients is the glycolysis pathway enrichment, which has been linked to greater SRAS-Cov-2 activity. Due to the TLR involvement and subsequent activation of the PI3K/Akt pathway, the primary energy metabolism shifts from lipid to glycolysis to create ATP amidst viral and bacterial infections, particularly following macrophage polarization and dendritic cell activation [49].

Cytokines such as IL-6, TNF, and IL-1 have a vital role in the down-regulation of crucial physiological coagulant pathways, as well as the hyper-activation of platelets that can result in an aberrant clot formation [50]. TLR activation mediated by a COVID infection is responsible for the initiation of extrinsic coagulation pathways and inflammatory signaling in endothelial cells, leukocyte infiltration, neutrophil granule leakage, platelet aggregation, and fibrin deposition that cause thrombosis [51]. Thrombosis caused by a cytokine storm, antiphospholipid antibody syndrome, macrophage activation syndrome, complement cascade, and RAS dysregulation are only a few of the thrombogenic processes that may be involved in the COVID-19 thrombosis. Although cytokine storm and IL-6, in particular, are substantially increased in COVID-19 patients compared to other septic etiologies, and these are mechanistically upstream of numerous thrombogenic pathways, it is currently unclear which set of pathways are prominent in COVID-19. Further studies are needed to better understand how these pathways could potentially help us develop novel, pathway-guided treatment strategies for the disease [52].

## 5. Conclusions

Our findings confirm that coagulation cascade dysregulation and subsequent disorders are common in COVID-19 infections and that thrombosis risk increases with chronic disease, old age, a positive D-dimer, unconsciousness, an increased coagulation profile, thrombocytopenia, and smoking. In addition, coagulation disorders cause intravascular clots in deep veins and in the pulmonary, cerebral, and cardiac circulations. Considering the high prevalence of coagulation disorders in COVID-19 patients in Al-Jouf, a national strategy for prevention and management is urgently needed and could provide public education programs for awareness focused on prevention of transmission modes, spread risk factors, signs, and symptoms. 

## Figures and Tables

**Figure 1 diagnostics-13-01085-f001:**
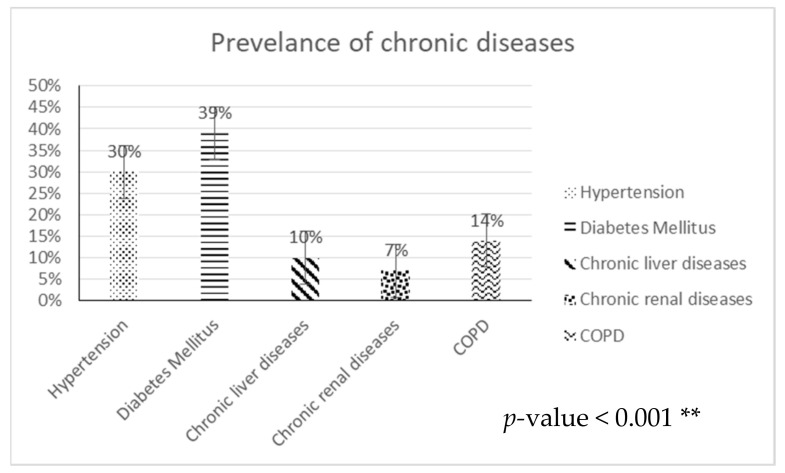
Chronic disease prevalence in COVID-19 patients (36% of the total number of COVID-19 patients).

**Figure 2 diagnostics-13-01085-f002:**
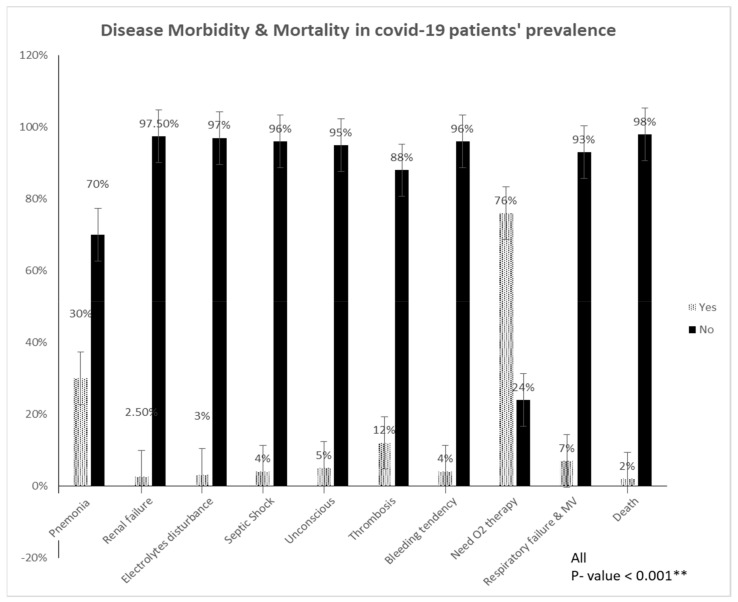
Disease morbidity and mortality in COVID-19 patients’ prevalence.

**Figure 3 diagnostics-13-01085-f003:**
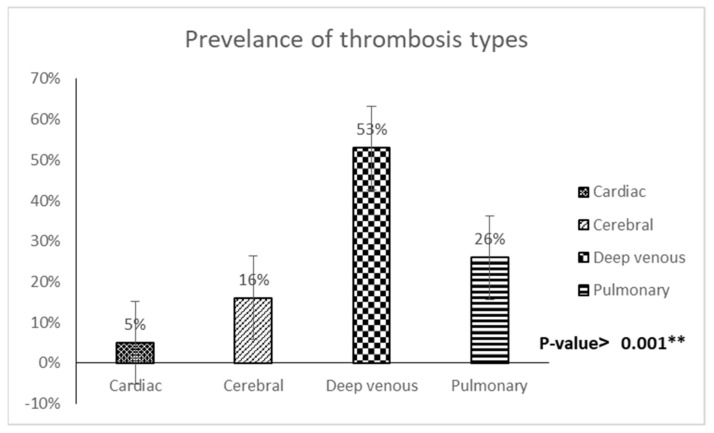
Types of thrombosis in COVID-19 patients with a prevalence complicated by thrombosis.

**Table 1 diagnostics-13-01085-t001:** Sociodemographic data features, patient hospitalization, and risk factors in COVID-19 patients.

Variable	Gender		Age		Patient Hospitalization		Risk Factors			
	M	F	19–59 years	60 years or More	Isolator	ICU	Smokers	Non-Smokers	Chronic Diseases	No Chronic Diseases
Prevalence	65%	35%	71%	29%	79%	21%	28%	72%	36%	64%
*p*-value	<0.001 **		<0.001 **		<0.001 **		<0.001 **		<0.001 **	

** means *p*-value is highly significant, <0.001.

**Table 2 diagnostics-13-01085-t002:** COVID-19 patients are sub-categorized according to platelet count, coagulation profile, and D-dimer results.

Variable	Normal	High	Low	*p*-Value
Platelet count	74%	8%	18%	<0.001 **
PT	55%	34%	11%	<0.001 **
aPTT	64%	28%	8%	<0.001 **
INR	55%	34%	11%	<0.001 **
D-dimer	Negative	Positive		*p*-value
	71%	29%		<0.001 **

** means *p*-value is highly significant, <0.001.

**Table 3 diagnostics-13-01085-t003:** Comparison of platelet count, coagulation profile, and D-dimer results in the control group and COVID-19 patients using an independent T test.

Variable	Control Mean ± SD	COVID-19 Mean ± SD	t	*p*-Value
Platelet Count (10^3^/mL)	323.5 ± 50.3	227.2 ± 95.1	3.6	0.001 *
PT (seconds)	11.9 ± 0.57	13.6 ± 1.3	−4.6	<0.001 **
aPTT (seconds)	28 ± 2.03	34.4 ± 6.1	−3.9	<0.001 **
INR	0.89 ± 0.09	1.12 ± 0.92	−2.9	0.006 *
D-dimer (mg/L)	0.33 ± 0.096	0.79 ± 0.59	−3.02	0.005 *

* means *p*-value is significant, < 0.01. ** means *p*-value is highly significant, <0.001.

**Table 4 diagnostics-13-01085-t004:** Prevalence of the most common symptoms and signs in COVID-19 patients’ infections.

Variable	Yes	No	*p*-Value
Fever	56%	44%	0.11
Cough	64%	36%	0.001 **
Sore throat	19%	81%	<0.001 **
Chest pain	30%	70%	<0.001 **
Dysnea	29%	71%	<0.001 **
Epigastric pain	7%	93%	<0.001 **
Diarrhea	5%	95%	<0.001 **
Muscle ache	41%	59%	0.03 *

* means *p*-value is significant, < 0.01. ** means *p*-value is highly significant, <0.001.

**Table 5 diagnostics-13-01085-t005:** Symptoms and signs related to or precipitating thrombotic risk in COVID-19 patients’ prevalence.

Variable	Yes	No	*p*-Value
Symptoms and signs of deep vein thrombosis	22%	78%	<0.001 **
Symptoms and signs of pulmonary thrombosis	29%	71%	0.79
Symptoms and signs of cerebral thrombosis	4%	96%	<0.001 **
Symptoms and signs of urinary tract thrombosis	3%	97%	<0.001 **

** means *p*-value is highly significant, <0.001.

**Table 6 diagnostics-13-01085-t006:** Relative risk and odds ratio (95% confidence interval (CI)) for thrombosis risk factors in COVID-19 patients.

Variable	Odds Ratio	Relative Risk	*p*-Value Fisher’s Exact Test
Gender: male	1.19(0.4–3.3)	1.16(0.46–2.9)	0.8(ns)
Age: 60 years or more	12.77 (3.96–41.2)	9.02 (3.2–25.7)	<0.001 **
Smoking	4.88 (1.8–13.18)	3.86 (1.67–8.9)	0.002 *
ICU cases	2.46 (0.89–6.85)	2.16 (0.92–5.07)	0.13 (ns)
Chronic diseases	18.04 (3.65–82.4)	13.55 (3.2–57.17)	<0.001 **
Low platelet count	5.78 (2.08–16.06)	4.2 (1.9–9.5)	0.001 **
High PT	6.8 (2.3–20.2)	5.3 (2.03–14.07)	<0.001 **
High aPTT	5.6 (2.04–15.4)	4.38 (1.8–10.4)	0.001 **
High INR	6.8 (2.3–20.2)	5.3 (2.03–14.07)	<0.001 **
Positive D-dimer	9.16 (3.07–27.3)	6.7 (2.57–17.6)	<0.001 **
Pneumonia	0.4 (0.11–1.44)	0.44 (0.13–1.43)	0.19 (ns)
Renal failure	2.56 (0.25–25.9)	2.17 (0.38–12.5)	0.4 (ns)
Electrolyte disturbance	1.9 (0.2–17.98)	1.72 (0.28–10.5)	0.47 (ns)
Septic shock	3.2 (0.58–17. 8)	2.57 (0.73–9.01)	0.19 (ns)
Unconsciousness	8.5 (1.9–37.3)	4.75 (2.06–10.9)	0.009 **
Respiratory failure and mechanical ventilator	4.75 (1.25–18.02)	3.39 (1.37–8.4)	0.033 *
Dead cases	15.4 (1.3–179)	5.8 (2.3–14.4)	0.041 *

* means *p*-value is significant, < 0.01. ** means *p*-value is highly significant, <0.001.

## Data Availability

All data supporting the reported findings are available upon request from the first and second co-authors.

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
