# Peer review of "Study of Coagulation Disorders and the Prevalence of Their Related Symptoms among COVID-19 Patients in Al-Jouf Region, Saudi Arabia during the COVID-19 Pandemic"

_diagnostics, 2023, doi:10.3390/diagnostics13061085_

Round 1
Reviewer 1 Report
The authors conducted a retrospective study on 160 Covid-19 patients.
Data was collected from the medical records department of King Abdulaziz Specialist Hospital, Sakaka, Al-jouf, Saudi Arabia.
The introduction is well-written but the language needs a lot of improvement in the medical terms, e.g Cardiac stroke as nothing is called Cardiac stroke, as well as grammar.
The authors did many unneeded statistical tests between the groups.
They have just considered that the presence or the absence of the risk factor or the blood test is a factor upon which they can do statistics and compare groups.
They had better do it like any registry where the data could be presented as number and percentage only. Then they can compare their results to other studies, or they can sub-categorize their cohort into groups according to age or sex or risk factors like diabetes.
The solid evidence of coagulation is also not mentioned and vague as we do not depend on D-dimer as evidence of coagulation at all.
D-dimer is good sensitive but not specific for thrombosis. (Sensitivity for DVT is around 97% but specificity is only 35%)
So, i think that the authors should revise their methodology and the design of the study, and they should re-formulate the results according to the design of a registry.
Author Response
Dear Reviewer,
Reviewer 1
Comments and Suggestions for Authors
The authors conducted a retrospective study on 160 Covid-19 patients.
Data was collected from the medical records department of King Abdulaziz Specialist Hospital, Sakaka, Al-jouf, Saudi Arabia.
Comment 1: The introduction is well-written but the language needs a lot of improvement in the medical terms, e.g Cardiac stroke as nothing is called Cardiac stroke, as well as grammar.
Reoly: We thank the reviewer for the insightful comments that helped to improve the quality of our work.
The language has been improved regarding medical terms, grammar and others.
Comment 2: The authors did many unneeded statistical tests between the groups.
Reply:
This work aims to assess the prevalence of coagulation disorders and their related symptoms among covid-19 patients. Thus the statistical tests were done in details to illustrate sociodemographic data features, patient hospitalization & risk factors in covid-19 patients. As well as, to determine Platelet count, Coagulation profile & D-dimer in covid-19 patients. Prevalence of most common symptoms and signs & that precipitating thrombotic risk were also demonstrated. Finally, the results showed the relative risk for Thrombosis in covid-19 patients.
Comment 3: They have just considered that the presence or the absence of the risk factor or the blood test is a factor upon which they can do statistics and compare groups.
Reply: Thank you for the comment which has helped to strengthen the findings.
The statistical tests were made to determine the relative risk & odds ratio (95% Confidence interval CI) and Fisher's exact test for Thrombosis risk factors in covid-19 patients depending on the data gathered from covid-19 patients with already diagnosed with thrombosis and other covid-19 patients without thrombosis (Table 6).
Comment 4: They had better do it like any registry where the data could be presented as number and percentage only. Then they can compare their results to other studies, or they can sub-categorize their cohort into groups according to age or sex or risk factors like diabetes.
Reply: Thank you for the comment.
Data were presented already as percentage and the significance of the data were analysed using Chi-square test. (Table 1, 2, 4, 5 & Figure 1, 2 & 3)
Sub-categorizing the cohort into groups according to risk factors mainly (Platelet count, Coagulation profile & D-dimer results) was done by dividing them into normal, high and low as shown in Table 2.
A control group was also added for Platelet count, Coagulation profile & D-dimer results to compare the results using independent t-test as illustrated in Table 3
The solid evidence of coagulation is also not mentioned and vague as we do not depend on D-dimer as evidence of coagulation at all. D-dimer is good sensitive but not specific for thrombosis. (Sensitivity for DVT is around 97% but specificity is only 35%).
Data collected for disease morbidity illustrating that 12% of covid-19 patients have thrombosis (Figure 2) were confirmed in the patient files not only by results of D-dimer tests but also with further investigations of thrombosis using Duplex ultrasonography & MRI.
Additionally, other studies reported that coagulopathy was a common symptom of covid-19 infection, with an increase in D-dimer, fibrinogen, PT, aPTT and moderate thrombocytopenia 28. Additionally, Zhang et al discovered that elevated D-dimer levels above 2.0 mg/L might predict mortality with a sensitivity of 92.3% and a specificity of 83.3% 29.
Similarly, Poudel et al found that D-dimer levels exceeding 1.5 μg/ml predict mortality in covid-19 patients with high specificity and sensitivity 30.
- Guan, W. J.; Ni, Z. Y.; Hu, Y.; Liang, W. H.; Ou, C. Q.; He, J. X.; Liu, L.; Shan, H.; Lei, C. L.; Hui, D. S. C.; Du, B.; Li, L. J.; Zeng, G.; Yuen, K. Y.; Chen, R. C.; Tang, C. L.; Wang, T.; Chen, P. Y.; Xiang, J.; Li, S. Y.; Wang, J. L.; Liang, Z. J.; Peng, Y. X.; Wei, L.; Liu, Y.; Hu, Y. H.; Peng, P.; Wang, J. M.; Liu, J. Y.; Chen, Z.; Li, G.; Zheng, Z. J.; Qiu, S. Q.; Luo, J.; Ye, C. J.; Zhu, S. Y.; Zhong, N. S.; China Medical Treatment Expert Group for, C., Clinical Characteristics of Coronavirus Disease 2019 in China. N Engl J Med 2020, 382 (18), 1708-1720.
- Zhang, Q.; Xu, Q.; Chen, Y. Y.; Lou, L. X.; Che, L. H.; Li, X. H.; Sun, L. Y.; Bao, W. G.; Du, N., Clinical characteristics of 41 patients with pneumonia due to 2019 novel coronavirus disease (COVID-19) in Jilin, China. BMC Infect Dis 2020, 20 (1), 961.
- Poudel, A.; Poudel, Y.; Adhikari, A.; Aryal, B. B.; Dangol, D.; Bajracharya, T.; Maharjan, A.; Gautam, R., D-dimer as a biomarker for assessment of COVID-19 prognosis: D-dimer levels on admission and its role in predicting disease outcome in hospitalized patients with COVID-19. PLoS One 2021, 16 (8), e0256744.
Comment 4: So, i think that the authors should revise their methodology and the design of the study, and they should re-formulate the results according to the design of a registry.
Reply: Thank you for the comment which has helped to strengthen our results. The methodology and the design of the study were revised according to the comments of the reviewer. Some results were re-formulated (as addition of control group in Platelet count, Coagulation profile & D-dimer results to compare the results using independent t-test as illustrated in Table 3. Comparison of platelet count, Coagulation profile & D-dimer results in control group & covid-19 patients using independent T test. Some tables were changed into graphs for better presentation of the results as shown in Figure 2. Disease Morbidity & Mortality in covid-19 patients' prevalence.

Reviewer 2 Report
The Authors Ghanem et al., in the manuscript entitled Study of Coagulation Disorders and the Prevalence of Their Related Symptoms Among COVID-19 Patients in Aljouf Region, Saudi Arabia, During the COVID-19 Pandemic, show a retrospective observational study and highlight how the coagulation cascade is a crucial mechanism in COVID-19 disease.
The manuscript could be improved. I invite the authors to review the manuscript in general and correct the major and minor revisions.
Authors need to double-check a few sentences that maybe seem similar to others already published.
Major revisions:
1. Materials and Methods: Is the study retrospective observational with a study group and a control group, or is it only a descriptive retrospective study of a group of sick patients? If so, please point it out and argue well the limitations of a descriptive observational study of only one group.
If, instead, the authors want to add a control group, they could refer to other studies, compare the data with these, and perform both a univariate analysis for coagulation values and a multivariate analysis taking into account chronic diseases.
2. Graphs of statistical analysis are only two; please, if possible, increase the statistical analysis with other graphs.
Minor revision
1. Introduction: Please Increase the description of COVID-19 disease symptoms to include neurological ones and mention at least 3/4 important references. I recommend seeing the WHO guidelines.
2. Correct everywhere from the abstract to the conclusion to the word covid-19 in COVID-19
3. Remove geographic locations from keywords and insert other keywords more relevant to the manuscript.
4. Materials and methods: in addition to the data collection period, indicate which variant of the virus was most widely circulating in Saudi Arabia then.
5. In discussion: at point “Coagulopathies resulting from covid-19 infection are possibly caused by numerous underlying mechanisms,” the authors might also consider other possible mechanisms, including an aspect arising from the gut microbiome:
a. Davis, H.E., McCorkell, L., Vogel, J.M. et al. Long COVID: major findings, mechanisms and recommendations. Nat Rev Microbiol (2023). https://doi.org/10.1038/s41579-022-00846-2
b. Brogna, Carlo et al. “Toxin-like Peptides from the Bacterial Cultures Derived from Gut Microbiome Infected by SARS-CoV-2-New Data for a Possible Role in the Long COVID Pattern.” Biomedicines vol. 11,1 87. 29 Dec. 2022, doi:10.3390/biomedicines11010087
c. Liu Q, Mak JWY, Su Q, et alGut microbiota dynamics in a prospective cohort of patients with post-acute COVID-19 syndromeGut 2022;71:544-552.
d. Gerlach, Joachim et al. “The immune paradox of SARS-CoV-2: Lymphocytopenia and autoimmunity evoking features in COVID-19 and possible treatment modalities.” Reviews in medical virology, e2423. 2 Feb. 2023, doi:10.1002/rmv.2423
e. Asakura, Hidesaku, and Haruhiko Ogawa. “COVID-19-associated coagulopathy and disseminated intravascular coagulation.” International journal of hematology vol. 113,1 (2021): 45-57. doi:10.1007/s12185-020-03029-y
6. Recheck punctuation and how to write references as suggested by the journal guidelines. Recheck all abbreviations when mentioned for the first time should be described in the full wording.
Author Response
Dear Reviewer,
Comments and Suggestions for Authors
The Authors Ghanem et al., in the manuscript entitled Study of Coagulation Disorders and the Prevalence of Their Related Symptoms Among COVID-19 Patients in Aljouf Region, Saudi Arabia, During the COVID-19 Pandemic, show a retrospective observational study and highlight how the coagulation cascade is a crucial mechanism in COVID-19 disease.
The manuscript could be improved. I invite the authors to review the manuscript in general and correct the major and minor revisions.
Authors need to double-check a few sentences that maybe seem similar to others already published.
Major revisions:
Comment 1: Materials and Methods: Is the study retrospective observational with a study group and a control group, or is it only a descriptive retrospective study of a group of sick patients? If so, please point it out and argue well the limitations of a descriptive observational study of only one group.
If, instead, the authors want to add a control group, they could refer to other studies, compare the data with these, and perform both a univariate analysis for coagulation values and a multivariate analysis taking into account chronic diseases.
Reply: Thank you for the insightful comment.
A control group was added to the analysis for coagulation values, platelet counts and D-dimer results using data gathered from 160 normal cases without covid-19 infection. Thus was illustrated as highlighted in the material and methods. The analysis was made using independent t-test to detect the significance as shown in the newly added Table 3.
Comment 2: Graphs of statistical analysis are only two; please, if possible, increase the statistical analysis with other graphs.
Reply: We thank the reviewer for the insightful comments that helped to improve the quality of our work. Some tables were changed into graphs for better presentation of the results as shown in Figure 2. Disease Morbidity & Mortality in covid-19 patients' prevalence.
Minor revision
Comment 1: Introduction: Please Increase the description of COVID-19 disease symptoms to include neurological ones and mention at least 3/4 important references. I recommend seeing the WHO guidelines.
Reply: Thank you for the comment, which has helped to strengthen the introduction. We have added two paragraphs, No 2 and 3 in the introduction including references.
Comment 2: Correct everywhere from the abstract to the conclusion to the word covid-19 in COVID-19
Reply: Thank you for the comment. It has been done
Comment 3: Remove geographic locations from keywords and insert other keywords more relevant to the manuscript.
Reply: Thank you for the comment. They have been removed. Our new Keywords include: covid-19; coagulation disorders; D-dimer; SARS-CoV-2; symptoms
Comment 4: Materials and methods: in addition to the data collection period, indicate which variant of the virus was most widely circulating in Saudi Arabia then.
Reply: The Delta variant of Covid-19 was detected more in many researches e.g.
Alahmad A M, Kamel S A, Alsulimani S T, et al. (June 16, 2022) Types of Variants Among Increased Cases of COVID-19 in the Western Region of Saudi Arabia in June 2021. Cureus 14(6): e26016. doi:10.7759/cureus.26016.
Comment 5: In discussion: at point “Coagulopathies resulting from covid-19 infection are possibly caused by numerous underlying mechanisms,” the authors might also consider other possible mechanisms, including an aspect arising from the gut microbiome:
- Davis, H.E., McCorkell, L., Vogel, J.M. et al.Long COVID: major findings, mechanisms and recommendations. Nat Rev Microbiol (2023). https://doi.org/10.1038/s41579-022-00846-2
- Brogna, Carlo et al. “Toxin-like Peptides from the Bacterial Cultures Derived from Gut Microbiome Infected by SARS-CoV-2-New Data for a Possible Role in the Long COVID Pattern.” Biomedicines vol. 11,1 87. 29 Dec. 2022, doi:10.3390/biomedicines11010087
- Liu Q, Mak JWY, Su Q, et alGut microbiota dynamics in a prospective cohort of patients with post-acute COVID-19 syndromeGut 2022;71:544-552.
- Gerlach, Joachim et al. “The immune paradox of SARS-CoV-2: Lymphocytopenia and autoimmunity evoking features in COVID-19 and possible treatment modalities.” Reviews in medical virology, e2423. 2 Feb. 2023, doi:10.1002/rmv.2423
- Asakura, Hidesaku, and Haruhiko Ogawa. “COVID-19-associated coagulopathy and disseminated intravascular coagulation.” International journal of hematologyvol. 113,1 (2021): 45-57. doi:10.1007/s12185-020-03029-y
Reply: We thank the reviewer for the insightful comments that assisted to improve the quality of the paper. All suggestions have been considered, indicating to the above studies.
Comment 6: Recheck punctuation and how to write references as suggested by the journal guidelines. Recheck all abbreviations when mentioned for the first time should be described in the full wording
Reply: Thank you for the comment. The have been manuscript has been rechecked regarding punctuation and the references have also been done as suggested by the journal guidelines.

Round 2
Reviewer 2 Report
The authors corrected what was indicated